# Delivery and Safety of a Two-Dose Preventive Ebola Virus Disease Vaccine in Pregnant and Non-Pregnant Participants during an Outbreak in the Democratic Republic of the Congo

**DOI:** 10.3390/vaccines12080825

**Published:** 2024-07-23

**Authors:** Hugo Kavunga-Membo, Deborah Watson-Jones, Kambale Kasonia, Tansy Edwards, Anton Camacho, Grace Mambula, Darius Tetsa-Tata, Edward Man-Lik Choi, Soumah Aboubacar, Hannah Brindle, Chrissy Roberts, Daniela Manno, Benjamin Faguer, Zephyrin Mossoko, Pierre Mukadi, Michel Kakule, Benith Balingene, Esther Kaningu Mapendo, Rockyath Makarimi, Oumar Toure, Paul Campbell, Mathilde Mousset, Robert Nsaibirni, Ibrahim Seyni Ama, Kikongo Kambale Janvier, Babajide Keshinro, Badara Cissé, Mateus Kambale Sahani, John Johnson, Nicholas Connor, Shelley Lees, Nathalie Imbault, Cynthia Robinson, Rebecca F. Grais, Daniel G. Bausch, Jean Jacques Muyembe-Tamfum

**Affiliations:** 1Institut National de Recherche Biomédicale, Kinshasa P.O. Box 1192, Democratic Republic of the Congo; hugokavunga@gmail.com (H.K.-M.); zmossoko@yahoo.fr (Z.M.); kepha4@hotmail.com (P.M.); jjmuyembet@gmail.com (J.J.M.-T.); 2Faculty of Infectious and Tropical Diseases, London School of Hygiene &Tropical Medicine, London WC1E 7HT, UK; kambale.kasonia@lshtm.ac.uk (K.K.); darius.tetsatata@lshtm.ac.uk (D.T.-T.); edward.choi@lshtm.ac.uk (E.M.-L.C.); hannah.brindle@lshtm.ac.uk (H.B.); chrissy.roberts@lshtm.ac.uk (C.R.); daniela.manno@lshtm.ac.uk (D.M.); badara.cisse@iressef.org (B.C.); mateus-kambale.sahani@lshtm.ac.uk (M.K.S.); nicholas.connor@lshtm.ac.uk (N.C.); daniel.bausch@finddx.org (D.G.B.); 3Mwanza Intervention Trials Unit, National Institute for Medical Research, Mwanza P.O. Box 1462, Tanzania; 4School of Tropical Medicine and Global Health, Nagasaki University, Nagasaki 852-8131, Japan; tansy.edwards@lshtm.ac.uk; 5Faculty of Epidemiology & Population Health, London School of Hygiene & Tropical Medicine, London WC1E 7HT, UK; benjamin.faguer@icloud.com; 6Epicentre, 75019 Paris, France; anton.camacho@epicentre.msf.org (A.C.); grace.mambula@epicentre.msf.org (G.M.); soumahaboubacar82@yahoo.fr (S.A.); michelkahumba@yahoo.fr (M.K.); benqbali@gmail.com (B.B.); kaninguesthermap@gmail.com (E.K.M.); costoreau@gmail.com (R.M.); oumar.toure@epicentre.msf.org (O.T.); paul.campbell@epicentre.msf.org (P.C.); mathilde.mousset@epicentre.msf.org (M.M.); robert.nsaibirni@epicentre.msf.org (R.N.); wandeybaisa@gmail.com (I.S.A.); kikongokambale@gmail.com (K.K.J.); rebecca.grais@pasteur.fr (R.F.G.); 7Janssen Vaccines and Prevention, 2333 CN Leiden, The Netherlands; bkeshinr@its.jnj.com (B.K.); crobin29@its.jnj.com (C.R.); 8Médecins Sans Frontières, 75019 Paris, France; john.johnson@paris.msf.org; 9Faculty of Public Health & Policy, London School of Hygiene & Tropical Medicine, London WC1E 7HT, UK; shelley.lees@lshtm.ac.uk; 10Coalition for Epidemic Preparedness Innovations, 0277 Oslo, Norway; nathalie.imbault@cepi.net; 11Foundation for Innovative New Diagnostics (FIND), Campus Biotech Chemin des Mines 9, 1202 Geneva, Switzerland

**Keywords:** Ebola, outbreak, vaccine, Ad26.ZEBOV/MVA-BN-Filo, DRC, safety, pregnant women, general population

## Abstract

During the 2018–2020 Ebola virus disease (EVD) outbreak, residents in Goma, Democratic Republic of the Congo, were offered a two-dose prophylactic EVD vaccine. This was the first study to evaluate the safety of this vaccine in pregnant women. Adults, including pregnant women, and children aged ≥1 year old were offered the Ad26.ZEBOV (day 0; dose 1), MVA-BN-Filo (day 56; dose 2) EVD vaccine through an open-label clinical trial. In total, 20,408 participants, including 6635 (32.5%) children, received dose 1. Fewer than 1% of non-pregnant participants experienced a serious adverse event (SAE) following dose 1; one SAE was possibly related to the Ad26.ZEBOV vaccine. Of the 1221 pregnant women, 371 (30.4%) experienced an SAE, with caesarean section being the most common event. No SAEs in pregnant women were considered related to vaccination. Of 1169 pregnancies with a known outcome, 55 (4.7%) ended in a miscarriage, and 30 (2.6%) in a stillbirth. Eleven (1.0%) live births ended in early neonatal death, and five (0.4%) had a congenital abnormality. Overall, 188/891 (21.1%) were preterm births and 79/1032 (7.6%) had low birth weight. The uptake of the two-dose regimen was high: 15,328/20,408 (75.1%). The vaccine regimen was well-tolerated among the study participants, including pregnant women, although further data, ideally from controlled trials, are needed in this crucial group.

## 1. Introduction

Between 2018 and 2020, North Kivu, South Kivu, and Ituri provinces in the Democratic Republic of the Congo (DRC) experienced the largest ever Ebola virus disease (EVD) outbreak in the DRC, resulting in 3481 cases and 2299 deaths (case fatality 66%) [1]. Despite control strategies, including ring vaccination with the rVSV-ZEBOV-GP vaccine, which has been shown to be highly protective, EVD cases continued to emerge during 2019, in part because of difficulties in contract tracing, delays in case detection, a complex humanitarian emergency and distrust of the response to the outbreak measures [2]. This prompted calls for additional control measures, including other EVD vaccines, which could potentially be administered more broadly to populations at risk in the area. In response, we conducted a study (DRC-EB-001) to investigate the effectiveness and safety of the two-dose Ad26.ZEBOV, MVA-BN-Filo vaccine regimen (Janssen Pharmaceuticals, Beerse, Belgium) to prevent EVD in the context of this outbreak [3]. However, because the outbreak came under control before the study could be fully implemented, we were unable to achieve the sample size to measure effectiveness [3].

The safety and immunogenicity of the Ad26.ZEBOV, MVA-BN-Filo vaccine regimen had been established in previous clinical trials in Europe and sub-Saharan Africa in non-pregnant adults, HIV-infected individuals, and children [4,5,6]. However, the vaccine had not previously been evaluated in pregnant and breastfeeding women who, although at the same risk of Ebola virus exposure as other subpopulations, are at greater risk of severe EVD and adverse pregnancy outcomes [7,8,9,10,11]. Here, we report on the secondary aims related to safety that include data on pregnant women and infants.

## 2. Materials and Methods

### 2.1. Study Design and Participants

DRC-EB-001 was a population-based, single-arm, open-label study of the Ad26.ZEBOV, MVA-BN-Filo vaccine regimen in adults and children aged >1 year in North Kivu. Because of ethical concerns in the setting of an ongoing outbreak, inclusion of an unvaccinated control group was considered unethical by the DRC authorities. After choosing the areas in which to conduct the study, community engagement, and social mobilization, vaccination was offered to people living or working in Kahembe and Majengo health areas of Goma city. The populations in these health areas were considered at high risk of EVD due to frequent contact with individuals from the outbreak epicentre to the north. We initially planned to further expand activities closer to the epicentre, but the outbreak ended before this could be implemented.

The study started on 14 November 2019. Dose 1 vaccinations were stopped in February 2020 when EVD cases declined, and measuring vaccine effectiveness for the protection against EVD was no longer feasible. Dose 2 vaccinations were suspended on 10 April 2020 due to public health restrictions related to the COVID-19 pandemic. Dose 2 vaccinations resumed on 15 September 2020 and continued until 9 February 2021. The assessment of immunogenicity of a delayed second dose for the Ad26.ZEBOV, MVA-BN-Filo vaccine regimen in a subset of participants who received dose 2 after the recommended interval is presented in a companion manuscript.

The study methods, including inclusion and exclusion criteria, have been previously described and are briefly reviewed here [3]. Participants were asked for written informed consent, and illiterate individuals provided witnessed informed consent. Children under 18 years require parent/guardian consent. Children aged 12 to 17 years were also asked to provide written assent. Based on the balance of potential risks and benefits, it was agreed that pregnant and breastfeeding women should be eligible for inclusion in the study from the start. Participants with an acute illness (excluding minor illnesses, e.g., mild diarrhoea) or a temperature ≥38.0 °C at the vaccination visit or who were being treated for malaria or who received routine immunization with a live attenuated vaccine within the last 30 days were deferred and rescheduled for vaccination at a later date.

Ad26.ZEBOV (5 × 10^10^ vp) was given as the first dose on day 0 as a 0.5 mL intramuscular (i.m.) injection, followed by MVA-BN-Filo (1 × 10^8^ Inf U) i.m. as a second dose on day 56, with a recommended window of −14/+28 days [12]. However, participants who presented after 84 days post-dose 1 were still able to receive dose 2 while vaccine activities were continuing. Dose 2 was withheld if participants had laboratory-confirmed EVD, experienced anaphylaxis, or any other serious adverse events (SAE) considered to be at least possibly related to dose 1 or for any other safety concern.

Pregnant and breastfeeding women were identified through self-reporting. Non-mandatory urine HCG pregnancy tests (Guangzhou Wondfo Biotech Co. Ltd., Guangzhou, China) were offered to female participants at vaccination visits if they believed that they might be pregnant and/or if the last menstrual period (LMP) was more than one month prior to the day of vaccination.

### 2.2. Safety Assessments

The first 500 adults and the first 500 children planned to be enrolled were included in adult safety and children safety subsets. They/their parents were actively contacted by telephone calls in order to collect SAE data at 30 days (−7 days/+1 month) post-dose 2.

Women reporting pregnancies from the time of dose 1 vaccination to 30 days post-dose 2 were followed within three months post-delivery by telephone or in person, if necessary, to collect pregnancy outcome data. Two pregnancy subsets were established: subset 1 aimed to include the first 250 pregnant women (at any trimester) at the time of dose 1, and subset 2 aimed to include the first 250 women who became pregnant within 30 days of either dose (i.e., exposed to the vaccine during the first trimester). Pregnancy subset participants were contacted by telephone to collect SAE data at seven days (−3/+7 days) and 21 days (−6/+7 days) post-dose 1 and at seven days (−3/+7 days), 1 month (−7 days/+1 month), 3 months (+/−14 days) and 6 months (−14/+28 days) post-dose 2 (unless the delivery was before these time points). An infant subset comprised the first 100 babies born to pregnant participants; these infants were examined by a paediatrician approximately three months after delivery.

Other participants were seen only on vaccination days 0 (dose 1) and 56 (dose 2; −14/+28 days). Data on SAEs were actively collected at dose 2 visits and passively collected up to 30 days post-dose 2 through self-reporting, with participants calling an emergency number or presenting to a vaccination site or a designated healthcare centre. Information was provided on how to contact the study team and where to seek care if needed.

### 2.3. Data Management

Vaccination data were collected on electronic case report forms (eCRFs) on password-protected tablets using the Open Data Kit (ODK Collect v1.16 to v1.22 and ODK Aggregate v1.44 to v2.03; https://opendatakit.org) collect application and synchronised to a REDCap (version 11.0.3; the REDCap Consortium) mirror database. A separate database recorded the longitudinal follow-up of subset participants. SAEs, pregnancy, and infant outcome information were collected using paper case report forms (CRF) that were double-entered into REDCap databases. Data queries were generated using R version 4.1.0 (R Corp Team) and Stata version 17 software (StataCorp LLC, College Station, TX, USA) and sent for investigation and resolution.

### 2.4. Statistical Analysis

Statistical analyses were produced using Stata, version 17 (StataCorp LLC, College Station, TX, USA) and included consenting participants who received dose 1. Socio-demographic characteristics were summarised by each dose. All individuals receiving at least one vaccination were included in safety analyses.

SAE were included in analyses if they occurred up to one month post-dose 2 for non-pregnant participants and up to delivery in pregnant women if the SAE occurred more than 30 days post-dose 2. SAEs were classified according to the Medical Dictionary for Regulatory Activities (MedDRA) 23.1 preferred terms.

Pregnancy and neonatal outcomes were analysed for pregnancies reported anytime from receipt of dose 1 and with conception up to one-month post-dose 2. A pregnancy was considered exposed to a vaccine dose if the conception date was before vaccination, with conception date calculated as the date of last menstrual period (LMP) plus 14 days, where LMP data were available. Breastfeeding at the time of each dose was reported.

The incidence of SAEs was summarised by timing of occurrence in relation to each vaccine dose and inclusion in one of the subsets, overall, for those reporting a pregnancy and overall, for non-pregnant individuals (i.e., those never reporting a pregnancy during the study). No hypothesis testing of safety data was planned.

Pregnancy outcomes were described overall by subset and vaccine exposure. Outcomes of interest were miscarriage, stillbirth or live birth, twin pregnancies, vaginal delivery, caesarean section (CS) rates, and reasons for having a CS. Among babies, outcomes were preterm live births, low birthweight (LBW), congenital anomalies, and neonatal deaths (within 7 days and 28 days).

No comparisons were made across groups as these sub-groups did not represent independent comparable groups. Further details are provided in the Appendix A.

### 2.5. Ethics Review and Approval

The London School of Hygiene & Tropical Medicine (LSHTM) was the study sponsor. The study protocol was approved by the University of Kinshasa School of Public Health Ethics Committee, the Comité National d’Ethique de la Santé, the LSHTM ethics committee, and the Médecins Sans Frontières Ethics review board in October 2019. The study was registered on ClinicalTrials.gov (NCT04152486).

## 3. Results

### 3.1. Vaccinations

Of 20,723 people screened between 14 November 2019 and 29 February 2020, 20,427 (98.6%) received dose 1. Ineligibility reasons included declining to be followed up (N = 183), a history of allergies or contraindications to vaccination or recurrent generalised hives (N = 62), no informed consent (N = 47), aged < 1 year (N = 2), and a history of EVD (N = 2). Additionally, 19 dose 1 recipients were excluded from the analysis because their consent forms could not be located during archiving (Figure 1).

Of the remaining 20,408 (98.5%), 67.5% were adults (≥18 years old) and 51.1% were male (Table 1). The median age (IQR) of males and females was 24 (14–37) and 21 (13–34) years, respectively.

Overall, 9551 participants had received dose 2 by the time of suspension of study activities in April 2020 due to COVID-19 public health restrictions. Overall, 15,328/20,408 (75.1%) participants received the second dose following the resumption of dose 2 vaccinations in September 2020; 9280 (45.5%) were within the target window, 6043 (29.6%) were between 4 and 15 months after dose 1, and 5 (0.0%) were less than 42 days after dose 1 (Appendix A). The age and sex distributions of dose 2 recipients were similar to those for dose 1 (Table 1).

Of the 5080 participants who did not receive dose 2, 5070 were lost to follow-up (i.e., uncontactable or could not return to Goma because of COVID-19 travel restrictions), four had died, four had contraindications to further vaccination (two with previous hives, one with epistaxis and one who reported difficulty moving his left upper arm after dose 1 and was concerned about receiving another dose), one experienced a serious adverse reaction (SAR) (described below), and one withdrew consent.

### 3.2. Safety Results

There were 494 participants in the adult safety subset, 492 in the children safety subset, 272 in pregnancy subset 1, and 88 in pregnancy subset 2 (Appendix A).

#### 3.2.1. SAEs in Non-Pregnant Participants

Overall, 50/19,187 (0.3%) non-pregnant participants (12 adults and 38 children) reported one or more SAEs post-dose 1 (Table 2), with 62 SAEs in total (0–3 per person reporting). The most frequently reported SAEs were malaria and typhoid fever in adults (six cases each, Appendix A) and malaria and gastroenteritis in children (three and two cases, respectively).

One SAR was reported in a non-pregnant 21-year-old female treated for anaphylactic shock, with symptoms occurring within 15 min after dose 1 that were considered possibly related to the vaccine. She felt unwell, with low blood pressure (90/50 mmHg), tachycardia, and oxygen saturation of 65–70%. The treating physician did not report signs of hives, facial swelling, or difficulty breathing. The participant lost consciousness for a few minutes but responded to treatment with intramuscular adrenaline (1 mg), intravenous hydrocortisone (100 mg), and 500 mL Ringer’s Lactate. After observation in the hospital, she was discharged home on the same day. No other documented SAEs were considered related to the study vaccines.

Overall, of a total of 20,408 participants who received dose 1, six, including one pregnant woman (see below), had fatal SAEs (Appendix A). A baby born with a congenital abdominal wall anomaly and evisceration also died. None of these SAEs was deemed related to vaccination.

#### 3.2.2. Pregnancy Cohort

There were 1238 pregnancies in 1221 women (17 women reporting a second pregnancy during the course of the study, Figure 2). Gestational age was known for 1037 (83.8%) pregnancies (Appendix A), of whom 487 (47.0%) had exposure to dose 1 only, 258 (24.9%) had exposure to dose 2 only, and 244 (23.5%) had exposure to both doses (Appendix A). Forty-eight pregnancies were not exposed to either dose. The estimated conception date and gestational age at vaccination or SAE diagnosis (when an SAE occurred during pregnancy) could not be calculated for the remaining 201 pregnancies. Breastfeeding was reported by 1040 women at the dose 1 visit and by 920 women at the dose 2 visit (Appendix A).

#### 3.2.3. SAEs in Pregnant Women

Among the 1221 women reporting pregnancy during the study, 371 (30.4%) experienced at least one SAE, including 258 (69.5%) who had at least one caesarean section (CS) (Table 3, Appendix A). SAEs in pregnant women were mainly hospitalisations for pregnancy-related events, primarily CS. The main indications for the 260 recorded CSs were a previous uterine scar (e.g., due to a previous CS; 139; 53.5%), cephalo-pelvic disproportion (55; 21.2%), and foetal distress (39; 15.0%) (Appendix A).

Of 1221 pregnant women, 31 (2.5%) reported a stillbirth, and 15 (1.2%) experienced threatened labour (Table 3). Other SAEs occurred in less than 1% of the 1221 pregnant women and were comparable to those observed in non-pregnant participants.

#### 3.2.4. Pregnancy Outcomes

Outcome data were not available for 69/1238 (5.6%) pregnancies: 55 of these were lost to follow-up, 13 were subsequently denied by the participant during follow-up, and 1 woman died of shock of unknown aetiology 27 days after dose 2, when she was estimated to be eight weeks pregnant.

Pregnancy outcome data were available for 260/272 (95.6%) and 81/88 (92.0%) pregnancy subsets 1 and 2, respectively (Appendix A). Of the 1169 pregnancies with a known outcome, 55 (4.7%) ended in miscarriage and 30 (2.6%) were stillbirths. Of 1084 (92.7%) pregnancies with at least 1 baby born alive, 1067 were singleton pregnancies and 17 twin pregnancies, resulting in 1100 live births (1 twin pregnancy resulted in 1 live birth and 1 stillbirth) (Figure 2). Of the 1114 pregnancies resulting in a live or stillbirth, 260 (23.3%) were CSs (250 singleton and 10 twin pregnancies). A total of 2 of 17 women with more than one pregnancy had a CS for both pregnancies. One twin pregnancy ended in one vaginal delivery and one CS.

Overall, 79/1032 (7.7%) live births with a recorded birth weight were classified as LBW. Among 672 babies estimated by LMP date to be full-term at delivery (≥37 weeks of gestation), 33 (4.9%) had LBW. Among 891 babies whose mothers had an LMP date, 188 (21.1%) were classified as preterm (<37 weeks gestation). Neonatal deaths within seven days occurred in 11/1100 (1.0%) liveborn babies (Appendix A). Five babies had a congenital anomaly (Appendix A), none of which were deemed related to the study vaccines.

## 4. Discussion

In this study of more than 20,000 individuals, the Ad26.ZEBOV, MVA-BN-Filo vaccine regimen was well tolerated in adults, children, and pregnant women. Only a single SAR experienced after dose 1 in 20,408 total participants was considered possibly related to vaccination, suggesting that we can be 95% confident that the incidence of SARs after dose 1 is no more than 0.03%. This diagnosis was based on the assessment of the medical doctor attending the participant at the time of the event, but there was insufficient evidence to confirm the case definition of anaphylaxis according to the guidelines of the Brighton Collaboration Anaphylaxis Working Group [13]. Overall, 0.3% of non-pregnant adult and paediatric participants experienced an SAE within one month of the second dose. These results are consistent with previous studies, in which no safety concerns were seen with this vaccine in adults, adolescents, and young children [4,5,6,14,15,16].

The DRC-EB-001 study was the first study to offer this vaccine regimen to pregnant women, thus filling an important need in this group with high morbidity and mortality due to EVD. Since ethical considerations precluded the inclusion of an unvaccinated control group in our study, we reviewed the scientific literature as well as government records on pregnancy and neonatal outcomes in eastern DRC in order to have a comparator to assess the safety of the vaccine in pregnant women and their offspring. The full manuscript is under review, but key points are provided here to help interpret our findings.

Overall, 2% of vaccinated women who reported a pregnancy in our study experienced a stillbirth. This is not substantially dissimilar to the DRC National Health Information System (SNIS) data in 2020, which reported that 0.6% of deliveries in North Kivu ended with a stillbirth [17]. A recent study reported a stillbirth rate of 37.3–37.9 per 1000 births, with the highest rates in women with low socioeconomic status [18].

Of women ever pregnant in our study, 5% reported a miscarriage. Although data on miscarriage rates in the general population from this region are limited, the North Kivu SNIS data documented 3% of pregnant women experiencing a miscarriage in 2020, and one study reported that 17% of women in South Kivu who had a previous CS experienced a miscarriage [17,19]. The prevalence of LBW babies in our study was also similar to the 6% reported in the SNIS database [17]. We recorded a 0.5% rate of congenital anomalies, similar to the rate of congenital anomalies recorded in 0.4% of live births at a referral hospital in Ituri [20]. Unfortunately, SNIS data do not document congenital anomalies.

We estimated that 21% of babies were preterm, based on LMP recall data, compared to 5% of infants in a study of women admitted for CS in South Kivu [19] and 2% in the North Kivu SNIS database [17]. It is not clear why our study has a higher proportion of preterm babies, but this could be due to different study procedures for determining gestational age. LMP recall data, employed in our study, can misclassify preterm births, and ultrasound generally gives lower rates of preterm births [21,22,23]. Women who could not recall LMP dates in the general population in DRC can be referred for ultrasound, which may have led to lower preterm birth rates in the SNIS database.

Delivery by CS contributed to a high proportion of SAEs experienced in pregnant women; one in five pregnant women in our study had a CS. It is worth noting that our study occurred in an urban area and that data suggest that there is a higher CS rate in some DRC cities compared with rural areas. One study reported CS rates of 23% in urban Goma, 5% in Karisimbi (a semi-urban health zone), and 14% in Rutshuru (a rural health zone) [24]. In Rutshuru Health Zone, Médecins Sans Frontières organized exemption for CS user fees and ambulance services to address the unmet need for CS. Our study also provided an exemption for CS user fees, which could have increased the number of women undergoing a CS. A study at four referral hospitals in Goma found a CS frequency of 17% between 2019 and 2020, with a higher proportion of CS (34%) at the main provincial hospital that received referrals from outlying centres [25]. This study also reported similar indications for CS to our study, including uterine scarring, foetal distress, and dystocia. These data suggest that our study did not lead to an increase in CS rates compared to other deliveries in the city. In our study, 53% of the indications for undergoing a CS was having had a previous CS, which may have also contributed to the elevated CS rate.

Passive data collection for SAEs can lead to the under-reporting of adverse events. However, the SAE rate in those who were more actively followed through phone calls and/or visits was similar to those with less active follow-up. For example, 30% of women reporting pregnancy at any time in the study experienced an SAE compared to 31% of women enrolled in the dose 1 pregnancy subset, and both groups had similar rates of follow-up to their delivery outcomes. Similarly, the rate of SAEs was 0.3% in non-pregnant participants and 0.1% or less in the more intensively followed safety subset participants (Table 2). We cannot rule out under-reporting of SAEs after the COVID-19 outbreak began, although we did not observe a difference in rates of reported SAEs pre- and during the study pause.

We observed a good uptake for the second dose (75.1%) in this conflict-affected and resource-constrained setting, even with an extensive study pause (approximately 6 months). Community engagement activities were key and are especially critical in settings with a high level of suspicion of health and other government authorities [26,27,28,29,30]. Our findings suggest that the administration of a two-dose EVD vaccine is safe and would be feasible in future outbreaks in similar communities or in settings with fewer logistic and security challenges. However, it would be useful to determine the feasibility of administering a two-dose vaccine during EVD outbreaks in geographically remote areas or in areas with more mobile populations than Goma in the event that future outbreaks of EVD arise in such settings. The vaccine may also be appropriate for the prevention of infection in healthcare workers and others at high risk of Ebola virus exposure [31]. In 2020, the Ad26.ZEBOV, MVA-BN-Filo vaccine regimen was granted authorization by the European Medicines Agency following demonstration of safety in clinical trials and immunobridging data showing stimulation of an immune response, particularly the binding of antibodies to the Ebola virus surface glycoprotein in humans commensurate with those associated with protection in challenge studies in non-human primates [32,33,34].

Limitations of our study included the lack of a randomised control group for safety comparisons. In addition, funding constraints meant that it was not possible to enrol a prospective cohort of pregnant women from Goma as a non-randomised comparison group to measure maternal and birth outcomes. The absence of an unvaccinated control arm, therefore, makes the analysis of our safety results difficult to interpret, although rates of SAEs in non-pregnant participants were similar or lower than those seen in earlier clinical trials [6,15,16,35]. We recorded a higher rate of preterm births compared to one study and the national survey data, but this could be due to the fact that we followed up every pregnancy and were able to also diagnose moderate preterm birth. Further safety data on the vaccine regimen in pregnant women are warranted to confirm our results and will be available from the INGABO randomized controlled trial that is being conducted in pregnant women in Rwanda (clinicaltrials.gov NCT04556526) [36].

## 5. Conclusions

A two-dose prophylactic vaccine regimen for EVD was acceptable, well-tolerated, and safe when administered to adults, children, and pregnant women and can be delivered in an EVD-affected region with good uptake of both doses. Our results should be confirmed by further studies, ideally from controlled trials, in pregnant women.

## Figures and Tables

**Figure 1 vaccines-12-00825-f001:**
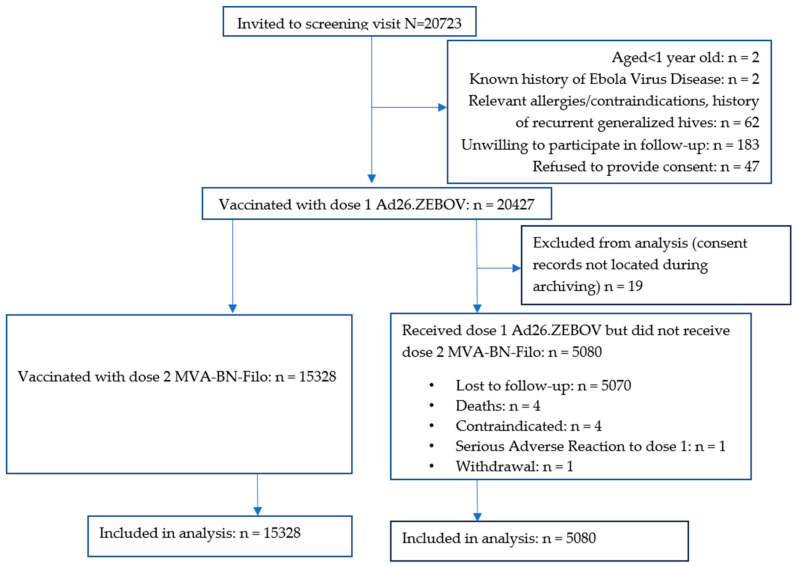
CONSORT Diagram. Participants′ vaccination flow.

**Figure 2 vaccines-12-00825-f002:**
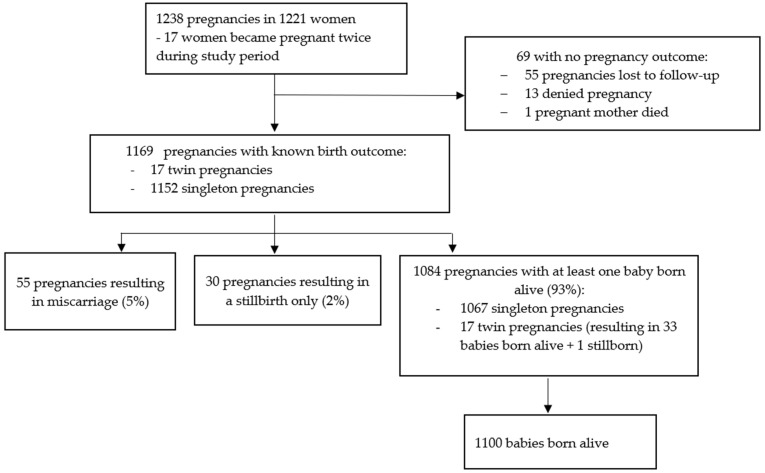
Pregnancy flow diagram.

**Table 1 vaccines-12-00825-t001:** Summary of participant characteristics at screening.

Number of Individuals Vaccinated		Dose 1 N = 20,408	Dose 2 N = 15,328
Vaccination site	1, n (%)	3834 (18.8)	2882 (18.8)
	2, n (%)	3161 (15.5)	2227 (14.5)
	3, n (%)	3420 (16.8)	2638 (17.2)
	4, n (%)	3358 (16.5)	2546 (16.6)
	5, n (%)	3608 (17.7)	2645 (17.3)
	6, n (%)	3027 (14.7)	2390 (15.6)
Received first vaccine at a site in health area of residence	Yes, n (%)	14,366 (70.4)	-
	No, n (%)	6037 (29.6)	-
	Unknown, n (%)	5 (<0.1)	-
Sex	Female, n (%)	9976 (48.9)	7704 (50.3)
	Male, n (%)	10,431 (51.1)	7624 (49.7)
	Missing, n (%) *	1 (<0.1)	0 (0)
Age (years)	65+, n (%)	503 (2.5)	405 (2.6)
	41–64, n (%)	3238 (15.9)	2622 (17.1)
	18–40, n (%)	10,031 (49.2)	7161 (46.7)
	12–17, n (%)	2240 (11.0)	1663 (10.8)
	4–11, n (%)	3212 (15.7)	2688 (17.5)
	1–3, n (%)	1183 (5.8)	789 (5.1)
	Missing, n (%) *	1 (<0.1)	0 (0)
MaleAge (years)	N	10,431	7624
	65+, n (%)	272 (2.6)	220 (2.9)
	41–64, n (%)	1835 (17.6)	1472 (19.3)
	18–40, n (%)	5199 (49.8)	3488 (45.8)
	12–17, n (%)	974 (9.3)	716 (9.4)
	4–11, n (%)	1554 (14.9)	1320 (17.3)
	1–3, n (%)	597 (5.7)	408 (5.4)
FemaleAge (years)	N	9976	7704
	65+, n (%)	231 (2.3)	185 (2.4)
	41–64, n (%)	1403 (14.1)	1150 (14.9)
	18–40, n (%)	4832 (48.4)	3673 (47.7)
	12–17, n (%)	1266 (12.7)	947 (12.3)
	4–11, n (%)	1658 (16.6)	1368 (17.8)
	1–3, n (%)	586 (5.9)	381 (4.9)
Awareness of the vaccination (N = 10,430 responses) †	Radio, n (%)	3388 (32.5)	-
	Poster, n (%)	1335 (12.8)	-
	Door-to-door, n (%)	6719 (64.4)	-
	Loudspeaker, n (%)	5439 (52.1)	-
	Leaflet, n (%)	467 (4.5)	-
	Other, n (%)	1997 (19.1)	-
Profession/child	≤15 years old, n (%)	5890 (28.9)	-
	Trader/Seller, n (%)	2539 (12.4)	
	Student (>15 years), n (%)	2036 (10.0)	
	Other manual worker ‡, n (%)	1317 (6.7)	
	Health worker §, n (%)	315 (1.5)	
	Skilled worker ‖, n (%)	663 (3.2)	
	Police/security, n (%)	143 (0.7)	
	Unemployed (>15 years)/Retired, n (%)	3794 (18.6)	
	Others ¶, n (%)	3711 (18.2)	

* Age and sex were not captured electronically for one participant. † Responses are not mutually exclusive (percentages sum to more than 100%). ‡ Here, other manual worker indicates driver, tailor, hairdresser, mechanic, and farmer. § Health worker definition includes doctors and nurses. ‖ Skilled worker stands here for teacher, technician, manager. ¶ All other professional categories.

**Table 2 vaccines-12-00825-t002:** Serious adverse events summary in non-pregnant participants.

	Adult Subset (≥18 Years) *	Child Subset (<18 Years) *	All Individuals Never Reporting a Pregnancy (Including Subsets)
Received dose 1, N	494	492	19,187
SAE during 15 min of dose 1, n (%)	0 (0)	0 (0)	1 (<0.1)
SAE up to 28 days post-dose 1, n (%)	2 (0.4)	3 (0.6)	28 (0.1)
SAE > 28 days post-dose 1, before dose 2, n (%)	0 (0)	0 (0)	11 (0.1)
SAE 28–112 days post-dose 1, no dose 2 †, n (%)	0 (0)	0 (0)	3 (<0.1)
Received dose 2 (at any time), N	420	449	14,340
SAE during 15 min of dose 1, n (%)	0 (0)	0 (0)	0 (0)
SAE up to 1-month post-dose 2, n (%)	1 (0.2)	0 (0)	8 (0.1)
SAE any time after dose 1, n (%) ‡	3 (0.6)	3 (0.6)	50 (0.3)
Total number of SAE §	3	3	62
Number of SAE per individual, median (range)	0 (0–1)	0 (0–1)	0 (0–3)

* The target size for each of the adult and child safety subsets was 500. The numbers in these subsets were reduced because of movement of female participants into the pregnancy subset upon notification of a pregnancy and because one individual in the adult and child subset was excluded as their informed consent document could not be located during archiving. † 112 days = 56 + 28 + 28, where 56 + 28 is the upper bound of the recommended window for dose 2 and the last 28 represents one extra month. ‡ Denominator is the number receiving dose 1. § Up to one-month post dose 2 or up to day 112 post dose 1 if no dose 2. It is important to note that some participants experienced more than one SAE; 62 SAEs comprise a total of 40 participants with 1 SAE, 8 participants with 2 SAEs, and 2 participants with 3 SAEs, 62 = 40 + 16 + 6.

**Table 3 vaccines-12-00825-t003:** Incidence of serious adverse events in pregnant women.

MedDRA Preferred Term by System Organ Class	Subset: Pregnant at Dose 1	Subset: Pregnant within 30 Days of Either Dose	Exposed to Dose 1 Only during Pregnancy	Exposed to Dose 2 Only during Pregnancy	Exposed to Both Doses during Pregnancy	All Pregnant Women, Including Subsets *
Number of participants vaccinated	272	88	487	257	244	1221
Congenital, familial, and genetic disorders						
Cleft lip	0 (0)	0 (0)	0 (0)	0 (0)	1 (0.4)	1 (<0.1)
Congenital tongue anomaly	1 (0.4)	0 (0)	1 (0.2)	0 (0)	0 (0)	1 (<0.1)
Exomphalos	0 (0)	0 (0)	1 (0.2)	0 (0)	0 (0)	1 (<0.1)
Gastrointestinal disorders						
Inguinal hernia	1 (0.4)	0 (0)	1 (0.2)	0 (0)	0 (0)	1 (<0.1)
Umbilical hernia	1 (0.4)	0 (0)	0 (0)	0 (0)	1 (0.4)	1 (<0.1)
Infections and infestations						
Appendicitis	1 (0.4)	0 (0)	1 (0.2)	0 (0)	0 (0)	1 (<0.1)
Ear infection	1 (0.4)	0 (0)	0 (0)	0 (0)	1 (0.4)	1 (<0.1)
Genitourinary tract infection	0 (0)	1 (1.1)	0 (0)	1 (0.4)	0 (0)	2 (0.2)
Malaria	2 (0.7)	0 (0)	2 (0.4)	1 (0.4)	2 (0.8)	4 (0.3)
Peritonitis	0 (0)	0 (0)	1 (0.2)	0 (0)	0 (0)	1 (<0.1)
Typhoid fever	1 (0.4)	0 (0)	0 (0)	0 (0)	1 (0.4)	2 (0.2)
Urinary tract infection	0 (0)	0 (0)	2 (0.4)	0 (0)	0 (0)	3 (0.2)
Urinary tract infection fungal	0 (0)	0 (0)	1 (0.2)	1 (0.4)	0 (0)	1 (<0.1)
Urogenital infection bacterial	1 (0.4)	0 (0)	0 (0)	0 (0)	1 (0.4)	2 (0.2)
Urogenital infection fungal	0 (0)	2 (2.3)	0 (0)	4 (1.6)	0 (0)	7 (0.6)
Injury, poisoning, and procedural complications						
Perineal injury	0 (0)	0 (0)	0 (0)	0 (0)	0 (0)	1 (<0.1)
Post-procedural complication	0 (0)	0 (0)	1 (0.2)	0 (0)	0 (0)	1 (<0.1)
Nervous system disorders						
Sciatica	0 (0)	0 (0)	0 (0)	0 (0)	1 (0.4)	1 (<0.1)
Pregnancy, puerperium, and perinatal conditions						
Abortion threatened	2 (0.7)	2 (2.3)	2 (0.4)	4 (1.6)	2 (0.8)	8 (0.7)
Abortion †	8 (2.9)	8 (9.1)	20 (4.1)	17 (6.6)	2 (0.8)	51 (4.2)
Anembryonic gestation	0 (0)	0 (0)	0 (0)	1 (0.4)	0 (0)	2 (0.2)
Ectopic pregnancy	0 (0)	2 (2.3)	0 (0)	2 (0.8)	0 (0)	2 (0.2)
Foetal death	3 (1.1)	0 (0)	11 (2.3)	1 (0.4)	3 (1.2)	18 (1.5)
Perineal injury	0 (0)	0 (0)	1 (0.2)	0 (0)	1 (0.4)	2 (0.2)
Postpartum haemorrhage	0 (0)	0 (0)	0 (0)	1 (0.4)	0 (0)	1 (<0.1)
Postpartum sepsis	0 (0)	0 (0)	0 (0)	0 (0)	0 (0)	1 (<0.1)
Premature delivery	2 (0.7)	0 (0)	4 (0.8)	0 (0)	0 (0)	7 (0.6)
Ruptured ectopic pregnancy	0 (0)	0 (0)	1 (0.2)	0 (0)	0 (0)	1 (<0.1)
Stillbirth	1 (0.4)	0 (0)	4 (0.8)	2 (0.8)	2 (0.8)	8 (0.7)
Threatened labour	4 (1.5)	2 (2.3)	5 (1.0)	2 (0.8)	3 (1.2)	15 (1.2)
Uterine cervical laceration	0 (0)	0 (0)	1 (0.2)	0 (0)	0 (0)	1 (<0.1)
Reproductive system and breast disorders						
Dysmenorrhoea	0 (0)	0 (0)	0 (0)	1 (0.4)	0 (0)	1 (<0.1)
Surgical and medical procedures						
Caesarean section	56 (20.6)	24 (27.3)	105 (21.6)	47 (18.3)	54 (22.1)	258 (21.1)
Vascular disorders						
Shock	0 (0)	1 (1.1)	0 (0)	1 (0.4)	0 (0)	1 (<0.1)
Shock haemorrhagic	0 (0)	0 (0)	1 (0.2)	0 (0)	0 (0)	1 (<0.1)

* Both the pregnant women deemed exposed and the women who became pregnant but were characterised as not being exposed to any vaccine but had at least 1 SAE have been included in this column. Data are n (%) of subjects with 1 or more events of the same preferred term. † Abortion includes MedDRA preferred terms of Abortion, Abortion complete, Abortion early, Abortion incomplete, Abortion spontaneous, Abortion spontaneous incomplete. NOTE: SAEs that occurred as congenital anomalies in babies born to vaccinated mothers are summarised in Appendix A. NOTE: The number of women ever reporting a pregnancy during the study is the denominator for this table, and the number who had at least one caesarean section is 258 in this table, whereas in the pregnancy outcome table (Appendix A), the denominator is the number of pregnancies and the number of caesarean deliveries is 260. This is because two women each had two pregnancies with caesarean section deliveries. NOTE: This table shows 51 women who experienced abortion (miscarriage), and 55 pregnancies ended in miscarriage in the pregnancy outcome table. The difference is due to the 4 SAEs with preferred terms of either anembryonic gestation or ectopic pregnancy; the remaining 51 SAEs were coded as one of the Abortion terms listed above.

## Data Availability

The rights of study subjects and partners, the sharing of data between partners, and the transfer of data to external third parties are governed by the Data Sharing Agreement. De-identified participant-level data collected in this trial will be disseminated through a FAIR-compliant data repository, such as the LSHTM Data Compass (https://datacompass.lshtm.ac.uk/ (accessed on 17 July 2024)), from 6 to 60 months after the publication of the main trial results. Other study documents (e.g., full protocol, statistical codes, Statistical Analytical Plan, DSMB Charter) will be available on request to Deborah Watson-Jones (corresponding author, ORCID: 0000-0001-6247-1746), Tansy Edwards (study statistician, ORCID: 0000-0002-6110-014X) or Edward Choi (study coordinator, ORCID: 0000-0002-8148-120X). The SNIS data belong to the Ministry of Health of the Democratic Republic of the Congo. Access to the North Kivu SNIS database can be provided on reasonable request from the Chef de Division of the North Kivu Division de la Santé. SNIS data are not publicly available.

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
