# Peer review of "Delivery and Safety of a Two-Dose Preventive Ebola Virus Disease Vaccine in Pregnant and Non-Pregnant Participants during an Outbreak in the Democratic Republic of the Congo"

_vaccines, 2024, doi:10.3390/vaccines12080825_

Round 1

Reviewer 1 Report

Comments and Suggestions for Authors

The manuscript represents an important study of a now EMA-licensed Ebola vaccine in special populations to include children and pregnant women. This is a well-written manuscript describing the prospects for use of an Ebola vaccine in pregnant women. There were challenges in the design of the study, but these were challenges associated with conducting studies during an active outbreak and the discussion acknowledges it.

Major points

1.       Lines 45-47 require significant clarification. The statement implies that the VSV based vaccine was ineffective which isn’t the case based on clinical trials and subsequent WHO analyses of real-world effectiveness. There is also concern that the statement is somewhat misleading. Unless there is a mis-understanding, this reviewer’s understanding was that the clinical trial mentioned could NOT be conducted in areas active in the outbreak and ONLY in surrounding provinces. Thus immediate outbreak control still relied on the VSV based vaccine.

2.       The discussion includes a good summary of if/how the adverse events related to pregnant women (pre-term births, miscarriages, etc.) may compare to expected rates. For the clinical team in-country, this may have benefited from surveys in the area (although admittedly that would be well beyond the scope of the study) to understand rates of these issues in a similar population. It may be worth trying to see if there is any more specific data to which comparisons could be made, although it’s understood that may not be possible. The paragraph in lines 339-354 starts to address it.

3.       Lines 370-376 would benefit from clarifying that the vaccine is indeed licensed for use by the EMA and including some detail on how immunobridging was used to infer efficacy and how that may apply to populations such as healthcare workers in a pre-exposure prophylaxis scenario.

Reviewer 2 Report

Comments and Suggestions for Authors

This is a study report of a clinical trial that aimed to assess the safety of the Ad26.ZEBOV, MVA-BN-48 Filo vaccine. The authors have provided details on the criteria for enrollment and inclusion in data analysis, as well as the methodologies of vaccination and other procedural aspects.

However, one major criticism of this study design is the absence of a randomized control group for safety comparisons. The authors themselves acknowledge this limitation. In order to improve the manuscript, the following major comments should be addressed:

The authors conclude that the two-dose prophylactic vaccine regimen for Ebola virus disease (ED) was acceptable, well-tolerated, and safe when administered to adults, children, and pregnant women. However, the text and data reveal a 5% miscarriage rate and 21% preterm birth rate. Although the authors argue that the use of last menstrual period (LMP) recall in this study may lead to misclassification of preterm births, and that ultrasound generally results in lower rates of preterm births, it is not clear how many study participants fell into this category.

Furthermore, among the women who underwent caesarean section, 21.2% experienced Cephalo-Pelvic Disproportion, and there were instances of low birth weight, placenta praevia, and short interpregnancy interval, among others. It is important to compare these rates with those recorded in other "safe" vaccination regimes. If such comparative data are not available, it would be valuable to compare these rates with the prevalence of these symptoms in the general pregnant population.

Without such comparative data, the claim of safety in pregnant women is not well-supported and could have serious consequences for the target population in the future.
